# An Effective Method of Aerobic Capacity Development: Combined Training with Maximal Aerobic Speed and Small-Sided Games for Amateur Football Players

Cansel Arslanoglu [1], Gurkan Selim Celgin [1], Erkal Arslanoglu [1], Nevzat Demirci [2], Fatih Karakas [3], Erol Dogan [3], Erdem Cakaloglu [4], Fatma Nese Sahin [4] and Hamza Kucuk [3],*

1 Department of Faculty of Sport Sciences, Sinop University, 57000 Sinop, Türkiye; canseloglu@sinop.edu.tr (C.A.); gscelgin@sinop.edu.tr (G.S.C.); erkaloglu@sinop.edu.tr (E.A.)
2 Department of Faculty of Sport Sciences, Mersin University, 33343 Mersin, Türkiye; nevzatdemirci44@hotmail.com
3 Department of Yasar Dogu Faculty of Sport Sciences, Ondokuz Mayis University, 55200 Samsun, Türkiye; karakasf24@gmail.com (F.K.); erol.dogan@omu.edu.tr (E.D.)
4 Department of Faculty of Sport Sciences, Ankara University, 06100 Ankara, Türkiye; ecakaloglu@ankara.edu.tr (E.C.); nesesahin@ankara.edu.tr (F.N.S.)
* Correspondence: hamza.kucuk@omu.edu.tr

**Abstract:** This study aimed to investigate the effects of combined training with maximal aerobic speed and small-sided games on football players' aerobic capacity development. Methods: The football players were divided into three groups as a combined training group (n = 20) (Mean ± SD age 23.40 ± 2.92 yrs, BMI was 23.67 ± 1.59 kg/m$^2$, mass: 74.80 ± 5.46 kg, height: 177.73 ± 4.31 cm), maximal aerobic speed group (n = 20) (Mean ± SD age 23.93 ± 2.46 yrs, BMI was 23.32 ± 1.42 kg/m$^2$, mass: 72.66 ± 5.38 kg, height: 176.46 ± 4.99 cm) and regular training group (n = 20) (Mean ± SD age 24.80 ± 5.84 yrs, BMI was 22.87 ± 1.67 kg/m$^2$, mass: 73.06 ± 6.74 kg, height: 178.33 ± 7.98 cm). In addition to regular football training, maximal aerobic speed training with small-sided games was applied to the combined training group (CT) and only maximal aerobic speed training was applied to the maximal aerobic speed group (MAS) twice a week for 12 weeks. The normal training group (NT) continued their routine football training programme. All participants applied the Yo-Yo IR1 test in the pre-test and post-test of the study. As a result of normality tests, it was determined that the data showed normal distribution, and the ANOVA test and Tukey's multiple comparison test were used in the intergroup evaluation. Results: When the Yo-Yo IR1 Test pre-test and post-test results were analysed, maximal aerobic speed training with small-sided games (CT) and maximal aerobic speed (MAS) groups were significantly higher compared to the normal training (NT) group regarding training load, running distance, and VO2max value among the football players ($p = 0.001$). There was no difference in the normal training group ($p > 0.05$). As a result, it was determined that combined training with maximal aerobic speed, small-sided games, and only maximal aerobic speed effectively improved football players' aerobic capacity and general performance.

**Keywords:** small-sided games; maximal aerobic speed; aerobic capacity; amateur; football

## 1. Introduction

Football is a sport in which physical, technical, and cognitive abilities play an important role in different player positions, depending on the position and level of competition [1,2]. The physical demands of football are receiving increasing attention due to its high-intensity intermittent nature [3]. Studies indicate that footballers must have well-developed physical characteristics to perform complex activities such as endurance, agility, speed, strength, and power generation during the game [4–6]. Furthermore, studies have shown that the physical demands of professional football increase over time [7]. These

demands give an overview of the dynamics of football by providing a holistic view of the physical characteristics and cognitive and technical aspects that characterise the game of football. In football, it is essential to understand the physical and physiological aspects to optimise player performance. Studies have examined various aspects of this issue. Soylu [8] (2021) compared physical, physiological, and anthropometric characteristics according to age and playing position. Evaluation of footballers' physical and physiological responses in different playing positions has been a focus, emphasising the importance of these factors in training and competition [9]. The study of the demands of football has provided insight into its physical, physiological, and tactical requirements [10]. Endurance is recognised as a critical physical factor in football, helping to prevent performance declines due to fatigue during matches [11]. Improving the aerobic capacity of footballers is essential to meet football's demands and perform at a competitive level [12]. Studies have shown that aerobic endurance training is important and that a significant proportion of training time improves aerobic capacity [13–16].

The pre-season is the most critical period of the macrocycle, requiring four to twelve weeks to prepare for the match period [17,18]. Training during this period is more intense than during the match period [19,20]. It aims to improve pre-season football players' aerobic, anaerobic, power, and strength capacities [21,22]. Considering that technical and tactical training will also be carried out during this period, achieving high efficiency in a short time is essential. Repetition of high-intensity exercises is an important performance parameter for football players [23,24]. Coaches train at different intensities, durations, and distances in order to obtain optimum performance in the pre-season [25], high-intensity interval training (HIIT) with or without the ball (with bouts ranging from 10 s to 6 min) [26], repeated sprint ability (RSA) training [27], or small-sided games (SSG) with different rules, numbers of players, and field dimensions [28,29].

Maximal aerobic speed (MAS) training in football is a very important element in improving players' aerobic capacity and overall performance [19]. High-intensity training, such as speed endurance training, has been recognised as a key strategy for improving aerobic fitness in football [30,31]. High-intensity interval training has significantly improved professional footballers' sprint time and maximal aerobic speed [5]. Recent studies have applied individualised speed zones to effectively measure external training load and indirectly calculate athletes' MAS through increased field tests [25,32]. These methodologies highlight the importance of personalised training programmes to optimise MAS and overall aerobic performance in football players. Small-sided games (SSG) are a valuable tool in football training for developing aerobic capacity and provide many benefits for players' physical, technical, and tactical development. Studies support that SSGs effectively improve cardiovascular endurance, increase pass accuracy, and increase VO2max in football players [33–35].

Based on their background knowledge, coaches perform drills without the ball for aerobic and anaerobic capacity development [26]. However, there is evidence that SSG twice a week improves aerobic endurance [36–40]. Radziminski (2013) stated that eight weeks of SSG improved aerobic capacity more than interval training [10], while another study reported that both methods showed similar improvements in young footballers [36,38]. In terms of aerobic capacity development, HIIT and SSG showed similar results [36,41]. In addition, it was stated that SSG training was more enjoyable. Conversely, the rating of perceived exertion, heart rate, and blood lactate responses were similar in young football players [41]. It was stated that footballers' technical and tactical skills [10,42–44] would improve during SSG, and it would provide appropriate training opportunities for the match [37,45]. It can be stated that SSG is more effective for improving both aerobic capacity and technical and tactical skills in football.

It has been stated that game-based training improves performance parameters in young footballers [37,39,45]. However, studies on combined training with SSG are limited compared to running-based training [46–48]. Clemente & Sarmento (2021) emphasised that SSG and running-based training methods are effective and that combining SSGs with running-based training will increase performance by increasing internal and external loads [49].

This study aimed to investigate the effects of combined training with maximal aerobic speed and small-sided games on the development of aerobic capacity in football players.

## 2. Materials and Methods

### 2.1. Study Design

The research of pre–post-test design was conducted over 12 weeks, starting from the special preparation period after the completion of the general preparation period of 4 weeks and according to the results of the Yo-Yo IR1 test after the necessary consents were obtained, the participants were divided into three groups: combined training group (n = 20), maximal aerobic speed group (n = 20), and normal training group (n = 20). Training intensity regularly increased in two training groups and mixed zones of aerobic and anaerobic loadings. The combined training group received maximal aerobic speed training (from 80% up to 110%; 80% in the 1st–3rd-week interval; 90% in the 4th–6th-week interval; 100% in the 7th–9th-week interval; 110% in the 10th–12th-week interval) with 4v4 small field games in the form of free play without target (Figures 1 and 2). Only maximal aerobic speed (MAS) training was applied to the training group, in addition to regular football training for 45–60 min 2 days a week for 12 weeks (80% in the 1st–3rd-week interval, 90% in the 4th–6th-week interval, 100% in the 7th–9th-week interval, 110% in the 10th–12th-week interval) (Figure 2). In maximal aerobic training, the running distance of each athlete according to the loading intensity was determined according to the Yo-Yo Intermittent Recovery Level 1 Test result. General and special warm-up was performed for 15–20 min before each training. Training volume and number of sets and sessions were balanced between the groups. The normal training group continued their normal football training programme.

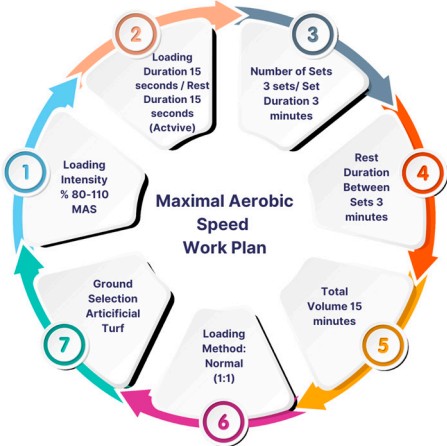

**Figure 1.** Maximal aerobic speed work plan.

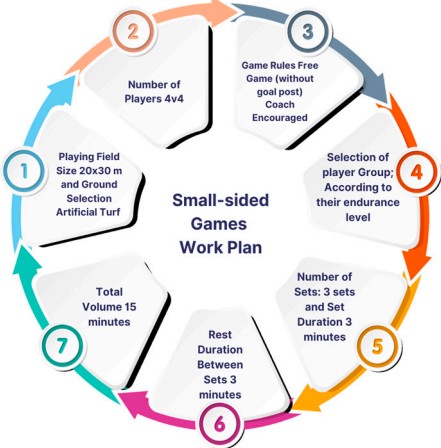

**Figure 2.** Small-sided games work plan.

Factors such as training intensity, exercise type, player numbers, rule changes, field dimensions, and coach support can be controlled during small-sided games. In their study, Rampinini et al. [50] (2007) examined the effects of the number of players, field dimensions, and coach support on training intensity. It has been observed that when coaches support the athletes during small-sided games, the training intensity is much higher in terms of heart rate, lactic acid, and perceived difficulty.

This study was approved by the Human Research Ethics Committee of Sinop University (approval number: E-57428665-050.04-253310) and conducted according to the Declaration of Helsinki. Informed consent forms were obtained from all participants.

### 2.2. Study Group

The study group consisted of 60 male football players (age mean CT: 23.40 ± 2.92; MAS: 23.32 ± 1.42; NT: 22.87 ± 1.67) who participated in amateur football training at amateur Football Clubs. Due to the pre-season period, they were at the same level in terms of health status, motivational factors, and physical conditions. Prior to the experimental training period, during the general preparation period, all subjects performed a comprehensive 4-week aerobic training programme of low severity in weeks 1 and 2 and, in weeks 3 and 4, subjects performed moderate severity activity (moderate and low-intensity running, dynamic stretching, and passing games, small-side games) at least 5 days a week for 60–90 min to adapt to training and prevent any injuries. As a result of the power analysis, it was determined that a sample size of 21 participants was sufficient (effect size: 0.50, confidence interval: 1-β 0.95, error probability: α 0.05, power value: 0.97). It appears that this study meets this suitability. All participants were informed about the research and their written consent was obtained.

### 2.3. Test Procedures

The footballers were informed that they should not train 24 h before the test day and that they should avoid stimulant drinks and foods such as alcohol and caffeine. Footballers were asked to come to the test rested and to have been fed at least 2–3 h before. The study's pre-test and post-test measurements were carried out simultaneously at Sinop University's football field (artificial turf). Athletes were asked to wear appropriate clothing for the tests. Participants were instructed to use their maximum capacity before the measurements. Before the measurements, it was checked whether the athletes had any health conditions that would prevent them from working. After 15 min of general warm-up and special warm-up (low-intensity running, dynamic stretching, and short passing), 3 min (jogging, stretching) recovery time was given and the test was started. After anthropometric measurements (height, weight, body mass index), a Yo-Yo Intermittent Recovery Level 1 Test was performed. At the end of the study, the same protocol was applied again for the post-tests.

#### 2.3.1. Anthropometric Measurements

The body height of the football players was measured with a Seca 213 brand height measuring device (Germany). Body weight was determined using Inbody 120 Bioimpedance body composition analyser and recorded in kg. The body mass indices of football players were determined using the formula "BMI $(kg/m^2)$ = Body Weight (kg)/Height$^2$ $(m^2)$".

#### 2.3.2. Yo-Yo Intermittent Recovery Level 1 Test

The "Yo-Yo 1 Intermittent Recovery Test" was used to determine the maximum oxygen utilisation capacity of the football players participating in the study. This test is an endurance test in which the test is performed in a $2 \times 20$ m area with gradually increasing speeds with a beep and a 10 s active recovery time consisting of $2 \times 5$ m. The test starts at a speed of 10 km/h and increases steadily. The test is terminated when the subjects reach the exhaustion point or do not reach the start and return lines three times in a row. Yo-Yo Intermittent Recovery Level 1 Test protocol was applied as specified in the literature [51–53].

Maximal Aerobic Speed

Maximal aerobic running speed was calculated using the Yo-Yo Intermittent Recovery Test Level 1 due to its suitability for the variable intensity movement structure of football. After the Yo-Yo test, the maximal aerobic speed parameters of the footballers were calculated using the formula established by Heaney et al.: MAS = $0.45625 \times$ YO-YO IR1 distance (km) + 3.61744 [54].

Calculation of VO2max Value of Athletes

Yo-Yo IR1 test: VO2max (ml/min/kg) = distance run (m) $\times$ 0.0084 + 36.4 [53].

Data Analysis

In this study, the SPSS 20.0 package programme was used for all statistical analyses. Descriptive statistics such as mean and standard deviation were used to evaluate the data. Shapiro–Wilk test was used to determine whether the data showed normal distribution and it was determined that the data showed normal distribution. In this context, the (one-way) ANOVA and Tukey's multiple comparison tests were used in the intergroup evaluation significance level was accepted as $p < 0.05$ in statistical analyses.

## 3. Results

Descriptive statistics of the participants are shown in Table 1.

**Table 1.** Descriptive statistics of subjects.

| Groups | CT Group | | MAS Group | | NT Group | |
|---|---|---|---|---|---|---|
| | n:20 | | n:20 | | n:20 | |
| Variable | Mean | SD | Mean | SD | Mean | SD |
| Sport Age | 10.60 | 3.15 | 12.60 | 6.76 | 11.75 | 5.30 |
| Age (year) | 23.40 | 2.92 | 23.93 | 2.46 | 24.80 | 5.84 |
| BMI (kg/m$^2$) | 23.67 | 1.59 | 23.32 | 1.42 | 22.87 | 1.67 |

CT = Combined Training; MAS = Maximal Aerobic Speed; NT = Normal Training.

When Table 2 was analysed, there was no significant difference when the pre-test results of the footballers were compared ($p > 0.05$; Figure 3).

**Table 2.** Intergroup comparison of participants' pre-test Yo-Yo IR1 endurance test values.

| | | **Distance Travelled (m)** | | | |
|---|---|---|---|---|---|
| Groups | n | Pre-Test | F | p | Cohen's d |
| | | Mean $\pm$ SD | | | |
| CT Group | 20 | 1618.66 $\pm$ 483.05 | | | |
| MAS Group | 20 | 1560.00 $\pm$ 479.52 | 0.35 | 0.707 | 0.036 |
| NT Group | 20 | 1485.33 $\pm$ 332.56 | | | |
| | | **VO2max** | | | |
| Groups | n | Pre-Test | F | p | Cohen's d |
| | | Mean $\pm$ SD | | | |
| CT Group | 20 | 50.00 $\pm$ 4.05 | | | |
| MAS Group | 20 | 49.51 $\pm$ 4.02 | 0.355 | 0.703 | 0.037 |
| NT Group | 20 | 48.88 $\pm$ 2.78 | | | |

CT = Combined Training; MAS = Maximal Aerobic Speed; NT = Normal Training.

When Table 3 was examined, it was determined that the CT and MAS groups were significant compared to the NT group in terms of training load, running distance and VO2max value among the football players ($p = 0.001$; Figure 4).

**Table 3.** Intergroup comparison of participants' pre-test and post-test Yo-Yo IR1 endurance test values.

| | | Running Distance (m) | | | | |
|---|---|---|---|---|---|---|
| **Groups** | **n** | **Pre-Test** | **Post-Test** | **F** | ***p*** | **Cohen's d** |
| | | **Mean ± SD** | **Mean ± SD** | | | |
| CT Group | 20 | 1618.66 ± 483.05 | 2677.33 ± 756.15 [a] | | | |
| MAS Group | 20 | 1560.00 ± 479.52 | 2674.66 ± 762.23 [a] | 8.920 | **0.001** | 0.004 |
| NT Group | 20 | 1485.33 ± 332.56 | 1746.66 ± 547.33 [b] | | | |
| | | VO2max | | | | |
| **Groups** | **n** | **Pre-Test** | **Post-Test** | **F** | ***p*** | **Cohen's d** |
| | | **Mean ± SD** | **Mean ± SD** | | | |
| CT Group | 20 | 50.00 ± 4.05 | 58.88 ± 6.36 [a] | | | |
| MAS Group | 20 | 49.51 ± 4.02 | 58.86 ± 6.40 [a] | 8.865 | **0.001** | 0.068 |
| NT Group | 20 | 48.88 ± 2.78 | 51.08 ± 4.60 [b] | | | |

CT-MAS > NT; CT = Combined Training; MAS = Maximal Aerobic Speed; NT = Normal Training.

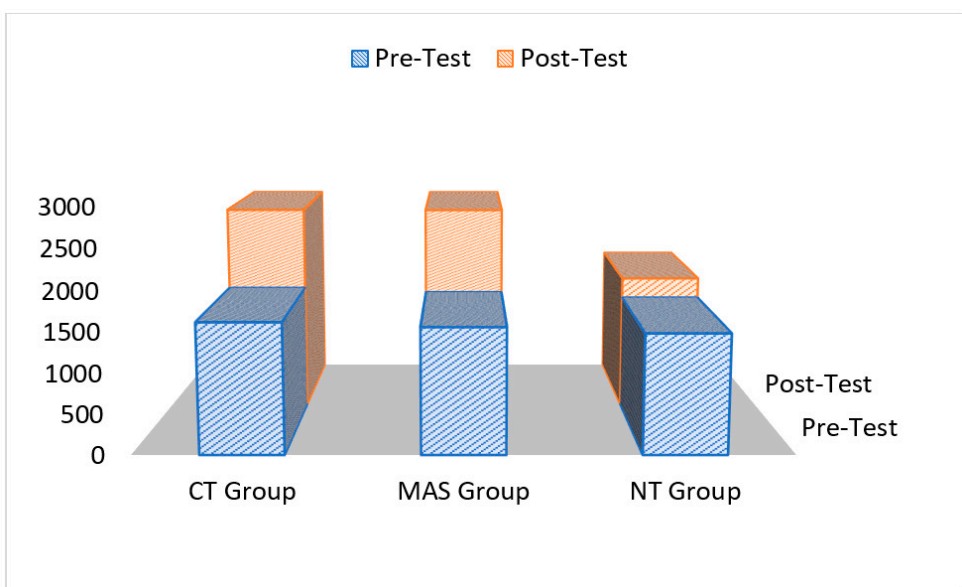

**Figure 3.** Pre- and post-test of running distance levels of the groups.

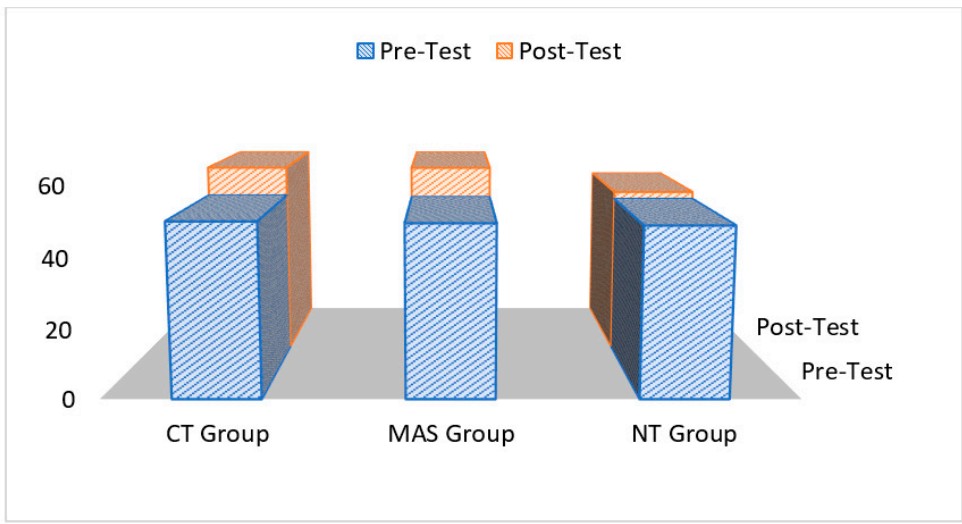

**Figure 4.** Pre-test and post-test of VO2 max levels of the groups.

## 4. Discussion

The hypothesis of this study proposed that a 12-week training programme that included twice-a-week, 4v4, and maximal aerobic speed training would improve the VO2max level and distance travelled of male football players. Our findings showed that the CT and MAS groups improved Vo2max and distance was covered better than the normal training group as a further improved endurance indicator.

Pre-season football training aims to prepare for the match period by increasing physical performance. In this training period, exercises that improve aerobic capacity are performed with technical and tactical skills [26,55,56]. Similarly, reducing the risk of injury during the match period is a process component [57]. Training small-sided games with maximal aerobic speed can lead to more fatigue during the season, depending on the intensity of the training. In this study, three different training programmes were applied to three different groups. The first group consisted of maximal aerobic speed and small-sided game combined (CT), the second group consisted of maximal aerobic speed (MAS), and the third group consisted of normal training (NT). The results showed that the CT and MAS groups developed more than the NT group. In addition, one of this study's important findings was that the distance covered in Yo-Yo IR1 progressed more in the CT and MAS groups than in the NT group ($p < 0.001$). The results obtained in the study agree with previous studies that pre-season training significantly improves aerobic capacity. In this regard, the time spent at high intensity should be 7–8% of the total training time during pre-season [58].

In our study, players in all groups have similar aerobic capacities and are moderately trained at baseline like in related studies [59–61]. However, the improvement observed in CT and MAS groups may be associated with the effect of high-intensity training with or without the ball. This development may be due to the high intensity of the footballers' training within the time allocated during the training period [37]. Thus, progressively increasing the load of the combined training (SSG + MAS) and maximal aerobic speed training programs (for 12 weeks) was more effective for inducing large improvements in aerobic capacity. The results show that players in combined training are more motivated during SSG under game-specific conditions than maximal aerobic speed and normal football training. SSG game demands an intensity close to match performance in terms of game format. Compared to interval training, this is a better alternative for football players [36,62].

Our findings are consistent with previous research investigating its effect on improving aerobic capacity. Maximal aerobic speed training provides valuable information about the effects on VO2max and aerobic capacity in footballers [13]. In the context of aerobic capacity and high-intensity training, Iaia et al. [54] (2009) emphasised the importance of repeated maximal velocities in developing aerobic endurance capacities in football players. Helgerud et al. [63] (2007) showed that high-intensity aerobic endurance training significantly improved VO2max. This suggests that HIIT may be effective in improving aerobic and anaerobic endurance, which are crucial for football performance [64]. In a study by Wong et al. [65] (2010), high-intensity interval training consisting of 15 s sprints was performed at 120% of the individual maximal aerobic rate with a 15 s rest. It has been shown that this training can be useful for improving aerobic endurance.

In another study conducted, four-week HIIT and MICT results showed increased anaerobic threshold, VO2 peak, and anaerobic power values. HIIT results were found to be more significant than MICT [32]. Furthermore, a meta-analysis by Clemente et al. [49] (2021) suggested that the internal load demands imposed by running-based long-interval HIIT were similar to those of small-sided games (SSGs), demonstrating the potential of HIIT to improve aerobic endurance in football players. In addition, the study by Fang et al. [66] (2021) highlighted the potential of HIIT to increase aerobic endurance in football players by repeating high-intensity training sessions lasting 5–45 s and recovery periods lasting 2–4 min for up to 40 min. It has been stated that HIIT applied at 120% of MAS allows a longer time to be spent at VO2max intensity compared to training with a continuous loading method applied at 100% of MAS [67]. In summary, it shows that high-intensity interval training methods can positively affect aerobic capacity in football players. It shows that HIIT

is an effective and valuable training method for improving aerobic endurance [68]. Small-sided games involving changes in the number of players, pitch size, and task constraints are effective in increasing aerobic endurance in football players [69]. In addition, these games have been found to improve aerobic capacity, strength, speed, and endurance in young footballers [10]. It has been shown that small-sided games can lead to increases in cardiovascular endurance, passing accuracy, and decision-making skills under pressure that closely resemble real match scenarios [34]. A systematic review has emphasised that small-sided games effectively incorporate specific football match demands and offer an efficient training solution [45,70]. Köklü et al. [71] (2015) compared the physiological responses of young football players in small-sided games and traditional aerobic running training and showed that small-sided games provided similar contributions to aerobic endurance. Arcos et al. [36] (2015) showed that small-sided games positively affected the aerobic endurance of young elite football players. In a study by Zaharia et al., 2023 [72], the effect of 4 vs. 4 small field games (SSG) with a goalkeeper (4 vs. 4 + SG) on the physical abilities of athletes performing specific aerobic endurance training was compared. It was found that the group applying the experimental training programme of 4 vs. 4 + GK SSGs significantly improved physical and functional parameters compared to the other training group. In another study, one group did higher aerobic training with a ball and strength training with plyometric and body-weight exercises. It was found that four weeks of high-intensity pre-season training improved aerobic capacity more than training with a ball [73]. These findings support the positive effect of small-sided games on aerobic capacity in football players and highlight its importance for improving physiological characteristics related to football performance.

Small-sided games and running training were equally effective in football players [38,74]. In a study conducted by Yüksel et al. [75] (2023), it was found that interval running training and small-sided games improved aerobic capacity in football players. Small-sided games increase aerobic capacity and various physical and motivational outcomes in football players [10,76]. The results of the literature support our study. It shows that the combined training (MAS and SSG) method and maximal aerobic speed method support aerobic performance development more.

There are some limitations in this study. The results obtained should be interpreted according to these limitations. The fact that the research group consists of amateur football players is one of the limitations of this study. The research was designed and the groups were randomly distributed. It is recommended that future studies should be conducted with different age groups and the evaluation of football players according to their positions in order to generalise the results. In addition, different designs should be added to the MAS, CT, and NT groups to evaluate the performance outputs.

## 5. Conclusions

As a result, it was determined that the combined training of maximal aerobic speed and small-sided games and only maximal aerobic speed training was an effective method to improve football players' aerobic capacity. Therefore, CT (SSG + MAS) and MAS training should be used during the pre-season period.

**Author Contributions:** Conceptualisation, C.A., G.S.C. and E.A.; methodology, C.A., G.S.C. and E.A.; software, N.D.; validation, F.K.; formal analysis, E.C. and E.D.; investigation, C.A., G.S.C. and E.A.; resources, F.N.S.; data curation, H.K.; writing—original draft preparation, C.A., G.S.C. and E.A.; writing—review and editing, C.A., G.S.C. and E.A. All authors have read and agreed to the published version of the manuscript.

**Funding:** This research received no external funding.

**Institutional Review Board Statement:** This study was approved by the Human Research Ethics Committee of Sinop University (approval number: E-57428665-050.04-253310) and conducted according to the Declaration of Helsinki. An informed consent form was obtained from all participants.

**Data Availability Statement:** The raw data supporting the conclusions of this article will be made available by the authors upon request.

**Acknowledgments:** All the authors are grateful for the participation of all those who wished to participate in this study.

**Conflicts of Interest:** The authors declare no conflicts of interest.

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
