# Peer review of "An Effective Method of Aerobic Capacity Development: Combined Training with Maximal Aerobic Speed and Small-Sided Games for Amateur Football Players"

_applsci, doi:10.3390/app14199134_

Round 1

Reviewer 1 Report

Comments and Suggestions for Authors

This study is of great interest and is fundamentally of practical application.

Introduction. When the importance of the aerobic component is mentioned, it should be clarified in which aspect, because the spectrum is very broad (capacity, power, lactic, alactic, etc.). As the description you make tries to be precise, it is preferable to provide these nuances.

As you indicate the importance of the distinction of roles, beyond indicating training means that could be used, you could give examples of the type of work that the bibliography indicates to be worked in each one.

In the title of your paper, you indicate that combined training will be discussed, but there is no explicit description in the introduction of what exactly it is. Please clarify this so that the reader knows what you are referring to.

When describing the sample, some data are missing; for example, the type of work they have done prior to this test is not the same from a period of inactivity to another period of conditioning, even if it is extensive.

However, there is no indication of the initial level of each player or of each of the differentiated groups. This initial level conditions the results and their interpretation.

The pre-test warm-up should be more detailed than indicating the duration of 15 minutes to calibrate pre-test levels (ventilatory, cardiac, etc.).

This could be added to the limitations of the study, not to other measuring instruments to serve as a contrast to the data reflected in the test.

It is not clear what combined training consists of and what loads and types of exercise it includes, as suggested in the title of the study.

Both the introduction and discussion indicate that HIIT is an effective method for soccer. Given that there are several types of HIIT, they could be more specific and could specify which types are the most appropriate.

I believe that they should include a section on the limitations of the study that would fill in some of the details that I personally miss: contrast measurement instruments, determination of the initial level of the sample, or comparison with women, which is not possible because it was not included.

Author Response

Comments 1: [Introduction. When the importance of the aerobic component is mentioned, it should be clarified in which aspect, because the spectrum is very broad (capacity, power, lactic, alactic, etc.). As the description you make tries to be precise, it is preferable to provide these nuances.]

Response 1: The importance of aerobic capacity was mentioned. – page number: 3, paragraph, and line: 97-99.

[Most coaches have traditionally used running drills without the ball to increase soccer players’ aerobic and anaerobic endurance as in many studies. However, it was proved that SSG twice a week in similar training improved aerobic capacity 36, 37, 38, 39, 40].

Comments 2: [As you indicate the importance of the distinction of roles, beyond indicating training means that could be used, you could give examples of the type of work that the bibliography indicates to be worked in each one.]

Response 2: Some changes were made in introduction related to training and  type of work. (Line 61-69)

Comments 3: [In the title of your paper, you indicate that combined training will be discussed, but there is no explicit description in the introduction of what exactly it is. Please clarify this so that the reader knows what you are referring to.]

Response 3: Thank you for pointing this out. We agree with this comment and some descriptions were added about HIIT, SSG and Combined training. – page number: 3, paragraph, and line: 100-121

[While one study found that SSG training enabled significantly greater progress in aerobic endurance than interval training after an 8-week intervention [10], some other studies showed that the two training methods had a similar influence on aerobic endurance in young soccer players [36, 38]. According to Selmi et al. [41] (2020) and Los Acros et al. [36], high intensity interval training (HIIT) and SSG sessions induced similar aerobic responses, while SSG induced a higher enjoyment level than HIIT. Furthermore, HIIT and SSG showed no significant difference in heart rate, rating of perceived exertion and blood lactate responses of youth soccer players [41]. Moreover, SSG as a game-based training improves aerobic and anaerobic performance just as much as running-based training. Furthermore, during SSG, players improve their performance levels, physical abilities [42, 43, 44], technical and tactical aspects [10] and experience similar situations to those they encounter in competitive matches [37, 45]. In conclusion, SSGs represent a versatile and effective training approach in football.Although many comparative studies have reported progress in performance parameters for game-based training [37, 45, 39] and MAS training [46] in youth soccer, to our knowledge, few studies have reported effects of combined training with SSG on performance parameters in young players [47, 48]. A recent systematic review [49] showed that combining SSGs and running-based training methods was effective. It showed that, compared to an intervention using solely SSGs, employing SSGs in combination with running-based training methods resulted in higher external and internal load values and greater increases in overall fitness capacity. Therefore, more research is needed to understand the efficiency of combined training for young players.]

Comments 4: [When describing the sample, some data are missing; for example, the type of work they have done prior to this test is not the same from a period of inactivity to another period of conditioning, even if it is extensive.]

Response 4: Information was given about the 4-weeks period before the study. – page number: 4, paragraph, and line: 159-166.

[The study group, consisted of 60 male football players to participate in amateur football training at amateur Football Clubs. Due to pre-season period, they were same level such as health status, motivational factors and physical conditions. Prior to the experimental training period, during the general preparation period all subjects performed a comprehensive 4-week aerobic training program consisting in weeks 1 and 2 of Low severity, in weeks 3 and 4 moderate severity activity (moderate and low-intensity running, dynamic stretching, and passing games, small-side games) at least 5 days a week for 60-90 minutes to adapt to training and prevent any injuries..]

Comments 5: [However, there is no indication of the initial level of each player or of each of the differentiated groups. This initial level conditions the results and their interpretation.]

Response 5: Thank you for this comment. Study group information revised according to comment. Sentence about participants was added. page number: 4, paragraph, and line: 158-165.

Comments 6: [The pre-test warm-up should be more detailed than indicating the duration of 15 minutes to calibrate pre-test levels (ventilatory, cardiac, etc.).]

Response 6: Before the test we were informed about the content of the warm-up program. – page number: 5, paragraph, and line: 180-182.

[After 15 minutes of warm-up general warm-up and special warm-up (low-intensity running, dynamic stretching, and short passing), 3 minutes (jogging, stretching) recovery time was given and the test was started.].

Comments 7: [This could be added to the limitations of the study, not to other measuring instruments to serve as a contrast to the data reflected in the test.]

Response 7: Thank you for raising this concern. A section on the limitations of the study has been added.– page number: 9, paragraph, and line: 327-334.

[This research poses some limitations; therefore, the findings should be interpreted with caution. First, research was designed as a amatuer players. Second, due to the research design, players were also randomly distributed among the groups. This research should be replicated with specific players’ positions from different age categories to generalize findings and to different challenge environments. Moreover, further studies should think of designing a training that can help investigators to see the influences of MAS, CT and NT on different performance parameters. In addition, different measurement tools should be utilised in future research.]

Comments 8: [It is not clear what combined training consists of and what loads and types of exercise it includes, as suggested in the title of the study.]

Response 8: Thanks for your attention. The intensity of the training and which zone  include and types of exercise were added. – page number: 3, paragraph, and line: 130-135.

[Training intensity regularly increased in two training groups and mixed zones of aerobic and anaerobic loadings. The combined training group received maximal aerobic speed training ( from 80% up to 110% (80% in the 1-3rd week interval, 90% in the 4-6th week interval, 100% in the 7-9th week interval, 110% in the 10-12th week interval) with 4v4 small field games in the form of free play without target (figure 1 and figure 2)]  

Comments 9: [Both the introduction and discussion indicate that HIIT is an effective method for soccer. Given that there are several types of HIIT, they could be more specific and could specify which types are the most appropriate.]

Response 9: Thank you for the comment, we revised the section to provide clear information. Information about HIIT and MICT was given. Also HIIT types were mentioned. – page number: 2, paragraph, and line: 61-69.

[Moderate-intensity continuous training (MICT) is commonly prescribed for the prevention and treatment of cardiometabolic diseases. However, this method is not specifically suited for addressing time constrains reported by physically inactive individuals. Alternatively, in the last decade, it has shown that high-intensity interval training (HIIT), which features short bouts of high-intensity exercise separated by periods of passive or active recovery at low intensity, induces similar or superior benefits on cardiometabolic health in healthy and clinical populations with less time commitment [15]. It seems that the intermittent periods of relatively intense exercise within a training session are responsible for these health benefits [16].

Comments 10: [I believe that they should include a section on the limitations of the study that would fill in some of the details that I personally miss: contrast measurement instruments, determination of the initial level of the sample, or comparison with women, which is not possible because it was not included.]

Response 10: Limitations of the study was added. – page number: 9, paragraph, and line: 327-334.

[This research poses some limitations; therefore, the findings should be interpreted with caution. First, research was designed as a amatuer players. Second, due to the research design, players were also randomly distributed among the groups. This research should be replicated with specific players’ positions from different age categories to generalize findings and to different challenge environments. Moreover, further studies should think of designing a training that can help investigators to see the influences of MAS, CT and NT on different performance parameters. In addition, different measurement tools should be utilised in future research].

Reviewer 2 Report

Comments and Suggestions for Authors

This study examines methods for aerobic capacity development in football athletes. The manuscript is generally well-structured. I only have a few comments that I hope the authors find useful. 

Lines 30-31: Please spell out CT, MAS, and NT on first use.

Line 32: Please maintain consistency in terminology. Use either "regular training group" or "normal training group" throughout the manuscript.

Figures 1 and 2: Please consider adding arrows to illustrate the sequence of the work/game plans if it flows in a specific direction.

Line 179: Please report effect sizes alongside statistical analysis results for a more appropriate/comprehensive understanding of the findings.

Lines 186-206: Given the significant F-test results, please consider conducting and reporting post-hoc analyses to identify specific group differences. Additionally, please address how you account for potential confounding variables such as participants' age, height, body weight, health conditions, and motivational factors.

Line 207: Please consider adding a limitations section to the discussion, addressing potential constraints of the study and maybe areas for future research. 

Author Response

Response 1: Explanations of abbreviations added. – page number: 1, paragraph, and line: 29-32.

[Results: When the Yo-Yo IR1 Test pre-test and post-test results were analysed, maximal aerobic speed training with small-sided games (CT) and maximal aerobic speed (MAS) groups were significantly compared to the normal training (NT) group regarding training load, running distance and VO2max value among the football players (p=0.001).]

Comments 2: [Line 32: Please maintain consistency in terminology. Use either "regular training group" or "normal training group" throughout the manuscript.]

Response 2: The phrases ‘normal training group’ were used throughout the manuscript. Consistency in terminology was maintained.

Comments 3: [Figures 1 and 2: Please consider adding arrows to illustrate the sequence of the work/game plans if it flows in a specific direction.]

Response 3: Figures 1 and 2: Revised. – page number: 4, paragraph, and line: 144-145.

        Figure 1. Maximal aerobic speed work plan                Figure 2. Small-sided games work plan

Comments 4: [Line 179: Please report effect sizes alongside statistical analysis results for a more appropriate/comprehensive understanding of the findings.]

Response 4: Cohen's d added for effect size to Table 2. – page number: 6, paragraph, and line: 219.

Distance travelled (m)

Groups

n

Pre-Test

F

p

Cohen’s d

Mean ± SD

CT Group

20

1618.66±483.05

0.35

0.707

0.036

MAS Group

20

1560.00±479.52

NT Group

20

1485.33±332.56

VO2max

Groups

n

Pre-Test

F

p

Cohen’s d

Mean ± SD

CT Group

20

50.00±4.05

0.355

0.703

0.0365

MAS Group

20

49.51±4.02

NT Group

20

48.88±2.78

Comments 5: [Lines 186-206: Given the significant F-test results, please consider conducting and reporting post-hoc analyses to identify specific group differences. Additionally, please address how you account for potential confounding variables such as participants' age, height, body weight, health conditions, and motivational factors.]

Response 5: Thank you for raising this concern. Therefore, we used the Tukey test for group differences. We mentioned about it in the data analysis. Study group information revised according to comment. Sentence about participants was added. page number: 4, paragraph, and line: 158-165.

[The study group, consisted of 60 male football players to participate in amateur football training at amateur Football Clubs. Due to pre-season period, they were same level such as health status, motivational factors and physical conditions. Prior to the experimental training period, during the general preparation period all subjects performed a comprehensive 4-week aerobic training program consisting in weeks 1 and 2 of Low severity, in weeks 3 and 4 moderate severity activity (moderate and low-intensity running, dynamic stretching, and passing games, small-side games) at least 5 days a week for 60-90 minutes to adapt to training and prevent any injuries.]

Comments 6: [Line 207: Please consider adding a limitations section to the discussion, addressing potential constraints of the study and maybe areas for future research.]

Response 6: Limitations and recommendations were added. – page number: 9, paragraph, and line: 326-333.

[This research poses some limitations; therefore, the findings should be interpreted with caution. First, research was designed as a amatuer players. Second, due to the research design, players were also randomly distributed among the groups. This research should be replicated with specific players’ positions from different age categories to generalize findings and to different challenge environments. Moreover, further studies should think of designing a training that can help investigators to see the influences of MAS, CT and NT on different performance parameters. In addition, different measurement tools should be utilised in future research.]  

Reviewer 3 Report

Comments and Suggestions for Authors

My recommendations are the following:

Title - I recommend mentioning that it is about amateur footballers.

Abstract: Lines 22-24 are repeated - maximal aerobic speed training, recommending correction and clarification. I recommend that when the 3 groups are presented, the acronyms for them should also be mentioned in parentheses, in order to corroborate in the Results section.

I recommend that body weight and height be deleted if BMI is presented.

Lines 29-31 recommend to mention descriptively what the acronyms represent, or their detailing according to the previous recommendation.

I recommend adding two more keywords.

 Lines 75-76 recommend using bibliographical sources.

Study design - I recommend mentioning the type of study.

Lines 189-194 recommend deletion, repeating the idea mentioned in table 1.

I recommend deleting figures 2 and 3 because the aspects mentioned in the tables are repeated. I recommend making an interpretation of the results, possibly referring to the differences in the testers/groups.

I recommend that HIIT and MICT should be mentioned descriptively.

At the end of the Discussion section, I recommend mentioning the limitations, practical implications and future research directions.

Lines 296-304 is an argument, I recommend moving it, without duplicating the information in the Discussion or Introduction section.

I recommend rewriting the Conclusions section based on the concrete findings from the present study.

Author Response

Response 1: The title revised. (Amateur added) – page number: 1, paragraph, and line: 1-4. [An effective method in aerobic capacity development: combined training with maximal aerobic speed and small-sided games in amateur football players].

Comments 2: [Abstract: Lines 22-24 are repeated - maximal aerobic speed training, recommending correction and clarification. I recommend that when the 3 groups are presented, the acronyms for them should also be mentioned in parentheses, in order to corroborate in the Results section.]

Response 2: Repeated sentences reorganised and abbreviations added. This section was revised. – page number: 1, paragraph, and line: 22-27/ 30-33

[In addition to regular football training, maximal aerobic speed training with small-sided games were applied to combined training group (CT) and only maximal aerobic speed training was applied to maximal aerobic speed group (MAS) twice a week for 12 weeks. The normal training group (NT) continued their routine football training programme. Results: When the Yo-Yo IR1 Test pre-test and post-test results were analysed, maximal aerobic speed training with small-sided games (CT) and maximal aerobic speed (MAS) groups were significantly compared to the normal training (NT) group regarding training load, running distance and VO2max value among the football players (p=0.001).]

Comments 3: [I recommend that body weight and height be deleted if BMI is presented.]

Response 3: Body weight and height removed in Table 1. - page number: 6, paragraph, and line: 220-221.

Groups

CT Group

MAS Group

NT Group

n:20

n:20

n:20

Variable

Mean

SD

Mean

SD

Mean

SD

Sport Age

10.60

3.15

12.60

6.76

11.75

5.30

Age (year)

23.40

2.92

23.93

2.46

24.80

5.84

BMI (kg/m²)

23.67

1.59

23.32

1.42

22.87

1.67

CT= Combined Training; MAS= Maximal Aerobic Speed; NT= Normal Training

Comments 4: [Lines 29-31 recommend to mention descriptively what the acronyms represent, or their detailing according to the previous recommendation.]

Response 4: Thank you for raising your concern, explanations of abbreviations revised and detailed according to recommendation. – page number: 4, paragraph, and line: 30-33.

[Results: When the Yo-Yo IR1 Test pre-test and post-test results were analysed, maximal aerobic speed training with small-sided games (CT) and maximal aerobic speed (MAS) groups were significantly compared to the normal training (NT) group regarding training load, running distance and VO2max value among the football players (p=0.001)]

Comments 5: [I recommend adding two more keywords.]

Response 5: Two more keywords added.

[Keywords: small-sided games, maximal aerobic speed, aerobic capacity, amateur, football]. [. – page number: 1, paragraph, and line: 36.]

Comments 6: [Lines 84-87 recommend using bibliographical sources.]

Response 6: Bibliographical sources added. . – page number: 2, paragraph, and line: 86-89. [Maximal Aerobic Speed (MAS) training in football is a very important element to improve players' aerobic capacity and overall performance [19]. High-intensity training, such as speed-endurance training, has been recognised as a key strategy for improving aerobic fitness in football [30,31].

Comments 7: [Study design - I recommend mentioning the type of study.]

Response 7: Thanks for this comment. Some cerrections were added about type of study. – page number: 3, paragraph, and line: 126-130.

[The research of pre-post-test design was conducted over 12 weeks, starting from the special preparation period after the completion of the general preparation period of 4 weeks and according to the results of the Yo-Yo IR1 test after the necessary consents were obtained, the participants were divided into three groups: combined training group (n=20), maximal aerobic speed group (n=20), and normal training group (n=20).].

Comments 8: [Lines 189-194 recommend deletion, repeating the idea mentioned in table 1.]

Response 8: Table 1. revised and removed according to comment 8. -page number: 6, paragraph, and line: 219.

Comments 9: [I recommend deleting figures 2 and 3 because the aspects mentioned in the tables are repeated. I recommend making an interpretation of the results, possibly referring to the differences in the testers/groups.]

Response 9: Figures removed. – page number: 4/5, paragraph, and line: 153/199.

Comments 10: [I recommend that HIIT and MICT should be mentioned descriptively.]

Response 10: Information about HIIT and MICT mentioned in the text according to recommendation.  – page number: 2, paragraph, and line: 61-69.

[Moderate-intensity continuous training (MICT) is commonly prescribed for the prevention and treatment of cardiometabolic diseases. However, this method is not specifically suited for addressing time constrains reported by physically inactive individuals. Alternatively, in the last decade, it has shown that high-intensity interval training (HIIT), which features short bouts of high-intensity exercise separated by periods of passive or active recovery at low intensity, induces similar or superior benefits on cardiometabolic health in healthy and clinical populations with less time commitment [15]. It seems that the intermittent periods of relatively intense exercise within a training session are responsible for these health benefits [16].  

Comments 11: [At the end of the Discussion section, I recommend mentioning the limitations, practical implications and future research directions.]

Response 11: A section added related to comment 11.– page number:9, paragraph,and line: 326-333.

[This research poses some limitations; therefore, the findings should be interpreted with caution. First, research was designed as a amatuer players. Second, due to the research design, players were also randomly distributed among the groups. This research should be replicated with specific players’ positions from different age categories to generalize findings and to different challenge environments. Moreover, further studies should think of designing a training that can help investigators to see the influences of MAS, CT and NT on different performance parameters. In addition, different measurement tools should be utilised in future research.].

Comments 12: [Lines 296-304 is an argument, I recommend moving it, without duplicating the information in the Discussion or Introduction section.]

Response 12: This argumant moved to discussion. – page number: 9, paragraph, and line: 319-325.

[Small-sided games and running training were equally effective in football players [74,38]. In a study conducted by Yüksel et al., [75] (2023), it was found that interval running training and small-sided game improved aerobic capacity in football players. Small-sided games increase aerobic capacity and various physical and motivational outcomes in football players [76, 10]. The results of the literature support our study. It shows that the combined training (MAS and SSG) method and maximal aerobic speed method support aerobic performance development more.]

Comments 13: [I recommend rewriting the Conclusions section based on the concrete findings from the present study.]

Response 13: Conclusion revised. – page number: 9, paragraph, and line: 335-338.]

[As a result, it was determined that the combined training with maximal aerobic speed and small-sided games and only maximal aerobic speed training were an effective method to improve football players' aerobic capacity. Therefore, CT (SSG+MAS) and MAS training should be used during the pre-season period.]  

Round 2

Reviewer 3 Report

Comments and Suggestions for Authors

no comments

Author Response

Dear reviewer, thank you for your support and interest.